# A process evaluation of an eHealth intervention to strengthen the circle of tuberculosis care in Shigatse, Tibet, China

Victoria Haldane[1], Zhitong Zhang[2], Tingting Yin[3], Bei Zhang[3], Yinlong Li[4], Qiuyu Pan[5], Katie N. Dainty[1], Elizabeth Rea[2], Pande Pasang[6], Jun Hu[6,7], Xiaolin Wei[2]*

**1** Institute of Health Policy, Management and Evaluation, University of Toronto, Toronto, Ontario, Canada, **2** Dalla Lana School of Public Health, University of Toronto, Toronto, Ontario, Canada, **3** Weifang Medical College, Weifang, China, **4** Jining Medical University, Jining, China, **5** North Sichuan Medical College, Nanchong, China, **6** Shigatse Centre for Disease Control and Prevention, Shigatse, China, **7** Shandong University of Traditional Chinese Medicine, Jinan, China

* xiaolin.wei@utoronto.ca

## Abstract

Tuberculosis (TB) is an ongoing global health threat that has been exacerbated by the COVID-19 pandemic. People with TB need comprehensive medical and social supports to ensure they can maintain and complete TB treatment. TB programs globally have turned to eHealth to bridge gaps in access and strengthen the circle of care around people with TB. This study evaluates the implementation of an intervention aimed at improving TB care in Shigatse, using the CFIR framework to identify factors influencing its success. The intervention included the use of e-Monitor boxes and WeChat for patient engagement. Data were collected through interviews with patients, treatment supporters, and health workers. Key challenges identified included inadequate infrastructure, digital literacy barriers, and unclear roles due to recent TB service delivery reforms. Enablers included strong social structures, proximity to village doctors, and government support for free TB treatment. Results showed that while older patients faced difficulties with digital tools, younger family members often assisted, enhancing engagement. Health workers' training and the timing of training sessions were critical to the intervention's success. The study concludes that despite challenges, the intervention was generally well-received and effective, with recommendations for ongoing training and adaptation to local contexts.

## Background

Tuberculosis (TB) remains a pressing global health threat, with disproportionate impacts in low- and middle-income countries (LMICs) [1]. Amidst ongoing health system strains, people with TB continue to face numerous barriers to accessing and maintaining treatment [2]. Adhering to lengthy treatment courses, however, is critical

**Data availability statement:** The datasets during and/or analysed during the current study are available from Shigatse CDC on reasonable request and after approval of the Ethics Review Committee of the Tibet Centre for Disease Control and Prevention. Data cannot be made publicly available for local ethical restrictions. The research team is not allowed to release the data without permission of Ethics Review Committee. Data requests should be sent to Shigatse CDC (shigatsecdc@163.com).

**Funding:** The trial is funded by TB REACH, Stop TB Partnership (to XW). The funder had no role in study design, data collection and analysis, decision to publish, or preparation of the manuscript.

**Competing interests:** The authors have declared that no competing interests exist.

to reduce morbidity and mortality from TB and to prevent antimicrobial resistance, which requires lengthier and more complex treatment regimens with poorer outcomes [3,4].

People with TB need comprehensive medical and social supports as they receive treatment. This 'circle of care' represents the patients' experience of receiving care and includes health care providers and family members or 'treatment supporters' who rally around the person with TB and provide both medical advice and psychosocial support [5]. The strength of this circle of care depends on many factors, but ongoing access to and engagement with trained health care providers is an important component. This is particularly important considering the gold standard for TB treatment remains Directly Observed Therapy (DOTS), which by nature requires observation by a health worker [6]. Yet, for many, access to health workers is elusive, and for TB programs, DOTS is difficult to fully implement, particularly in resource-constrained and remote settings, or as COVID-19 demonstrated, when in-person visits are not possible [7,8].

Given these challenges, TB programs globally have turned to eHealth to bridge gaps in access and strengthen the circle of care around people with TB [9,10]. One eHealth tool increasingly used are electronic monitoring boxes (e-Monitors) that track whether TB medications are taken as directed. Studies have reported varying degrees of success in their use, highlighting the need for more research on e-Monitor intervention effectiveness, cost, and implementation in resource-constrained and high-burden settings [11,12]. This work explores one such intervention in Shigatse, Tibet Autonomous Region (Tibet), China. A randomized controlled trial was undertaken to evaluate the effectiveness of an intervention to strengthen the circle of care around people with TB using a combination of e-Monitor boxes to measure treatment adherence, an app to connect people with their health care providers, and inclusion of family members as treatment supporters [13]. The trial found that using e-Monitors significantly improved medication adherence and treatment success among tuberculosis patients in Tibet. Patients with e-Monitors had a higher treatment success rate (94%) and better adherence compared to the control group [14].

It is important to match studies measuring effectiveness with studies to examine how the intervention works (or does not work), for whom, and why [15,16]. Implementation science offers a complementary approach to contextualize health service intervention and is an approach to untangle and explore the rich contextual factors, and adaptations, that shape intervention implementation [5,17,18]. To date few studies have gathered implementation insights on TB e-Monitor interventions implementation in China and none to our knowledge have explored the unique barriers and enablers present in Tibet [19–21]. Implementation experiences from Tibet can offer strategies to improve practice and policy in other resource constrained settings with a high burden of TB [22]. Guided by the Consolidated Framework for Implementation Research (CFIR), this process evaluation draws on interviews with patients, treatment supporters and health workers to identify challenges and enablers to intervention implementation and offer insights for ongoing program delivery. The CFIR framework was used because it provides a comprehensive and flexible structure to identify multilevel

factors, such as contextual, individual, and process-related elements, that influence the implementation of complex health interventions like eHealth for TB care in Shigatse. Implementation research outcomes such as sustainability were not directly measured. As such, this evaluation focuses on understanding contextual and process-related factors influencing implementation, rather than assessing specific implementation outcomes.

## Methods

### Ethics statement

Ethical approval for the study was obtained from the Office of Research Ethics at the University of Toronto (Ref: 36569) and the Ethics Review Committee of the Tibet Centre for Disease Control and Prevention (Ref: 006). Participants received information and consent forms in Chinese, translated into Tibetan if needed. Written informed consent was obtained from participants. All participants consented to audio recording and could refuse questions or withdraw at any time.

This study adhered to the consolidated criteria for reporting qualitative studies (COREQ). (S1 Checklist)

### Study setting and intervention

Shigatse is a remote and sparsely populated prefecture that is characterized by high altitude and rugged terrain. These impact access to care, shape people's ability to adhere to TB treatment and the region has a TB prevalence twice the Chinese national average [23–25]. The most recent data from Shigatse found that only 72% (769/1073) of new pulmonary TB cases completed treatment, with the majority, 83% (252/304), lost to follow-up [26]. Recent TB service delivery reforms have streamlined assessment and referral processes, while strengthening TB support at the most local level – township hospitals and village doctors [27]. As part of efforts to enhance the quality of TB care in Shigatse, an eHealth intervention was developed to better connect people with TB and their health care providers.

The intervention is comprised of three key components: 1) a Bluetooth-enabled medication monitor box (e-Monitor) linked to an online monitoring platform, 2) identification and onboarding of a family treatment supporter who receives education about TB and the importance of treatment adherence, and 3) communication using WeChat (a popular messaging application platform) between people with TB, their treatment supporter, and health care providers (TB doctor and village doctor). Using these components, the intervention offered a way to strengthen the circle of care around people with TB and support them through their treatment course. The intervention being assessed aims to enhance medication adherence and improve overall treatment outcomes amongst people newly diagnosed with pulmonary TB in Shigatse [14].

### Process evaluation framework

The trial and its complementary qualitative inquiry draws on the updated Consolidated Framework for Implementation Research, which encourages flexible application and offers numerous sub-constructs to help researchers probe implementation in ways that suit their research questions, innovation under study, and unique research context [17]. There is growing evidence of its use to guide study design and analysis of e-Health interventions in LMIC settings and increasing adoption of its constructs to guide process evaluations of TB interventions [28–31]. Table 1 offers the CFIR construct and its definition, as well as an overview of how we conceptualize these constructs in relation to the TB intervention being evaluated.

### Selection of participants, data collection, and processing

Participants included patients, their treatment supporters, and TB health workers or Centre for Disease Control staff. Patients were defined as those older than 15 years old, with presumed TB and newly confirmed pulmonary TB starting on standard 6-month short-course outpatient treatment. Family treatment supporters were those nominated family members who were participating in the intervention. Health worker participants included village doctors, township hospital doctors

**Table 1. CFIR framework constructs and study definitions.**

| Construct | Definition as related to our study |
|---|---|
| Outer setting:<br>*The setting in which the inner setting and innovation exist.* | Relates to broader contextual factors, including local conditions and attitudes determining patient access to TB care, that shape how people engage with the intervention. |
| Inner setting:<br>*The setting in which the innovation is implemented.* | Relates to health system organization and characteristics of implementation sites, as well as the how TB care is provided in Shigatse, Tibet. |
| Individuals:<br>*The roles and characteristics of individuals who engage with the innovation.* | Experiences, perspectives, and behaviours of people with TB, their treatment supporters, and health care providers that shape their experience of the intervention. |
| Innovation:<br>*The intervention being implemented.* | Features of the intervention that may influence its effective implementation in Shigatse, Tibet. |
| Implementation process:<br>*The activities and strategies used to implement the innovation.* | Description of how the TB intervention was implemented, this study both describes the process and is itself part of the process domain as it offers insights on innovation delivery. |

and nurses, county hospital doctors, as well as CDC staff working in Shigatse, Tibet. Participants were invited to the study by the research team affiliated with the randomized control trial of the intervention [13]. Purposeful and pragmatic recruitment was used that was sensitive to the resource constraints in the region (e.g., limited health workforce with little time, patients who traveled long distances or had other responsibilities), we also aimed for a balance of gender, age, health worker roles and location across four counties representing a mix of urban and rural locations. Participants were recruited from March 1, 2019 to December 31, 2020.

Researchers conducted interviews following a topic guide tailored to different participant categories. For patients and their supporters, topics included health status, experiences with the health system, TB care, medication-taking, technology use, and engagement with the intervention (e-Monitor box and WeChat). Health workers discussed their roles, TB care experiences, training, trial engagement, patient onboarding, and intervention components.

The study team included local and other researchers familiar with the study context. Interviews were held in private, comfortable locations. Interviews were conducted in Mandarin, with Tibetan interpretation as needed. Recordings were transcribed into Chinese and translated into English, with transcripts checked by a bilingual translator. More information can be found in our collaborative autoethnography describing the translation process [32].

Participant confidentiality in analysis and writing was ensured by using participant numbers and quoting participants using pseudonyms. Data was securely stored as per protocols defined by the University of Toronto Office of Research Ethics including password protection and use of secure drives.

The analysis was guided by a qualitative descriptive approach, which is often used in cross-cultural health services research given its emphasis on literal interpretation of translated narrative and limited movement 'into' the text [33–35]. Interviews were coded using framework analysis as described by Ritchie and Lewis and guided by the CFIR constructs to categorize our data [17,36]. NVivo 12 was used to organize the data and code transcripts. Two reviewers (VH, ZZ) reviewed the CFIR-guided coding framework based on a subset of interviews. After discussion of any discrepancies to reach agreement, the coding framework was applied to subsequent transcripts (S1 Fig). Interviews were stopped when no new concepts emerged in the final three interviews for patients and providers respectively, indicating thematic saturation had been reached based on our operational criterion [37].

## Results

A total of 62 interviews were conducted with people with TB, their treatment supporters, and health care providers. All approached participants agreed to be interviewed. The study was conducted in three rural counties (Gyangze, Sa'gya, and Tingri) and one urban district (Samzhubze). Interviewees were purposively sampled based on district to ensure representation from different settings and from different categories of stakeholders who engage with the intervention (Table 2).

**Table 2. Interview participants by category and study district.**

| Participant characteristics | n= |
|---|---|
| Patient | 20 |
| Family treatment supporter | 5 |
| Village doctor | 14 |
| Township hospital doctor/nurse | 14 |
| County hospital doctor | 7 |
| CDC staff | 2 |
| **District/county** | |
| Samzhubze | 16 |
| Sa'gya | 17 |
| Gyangze | 13 |
| Tingri | 16 |

We found that patients, their supporters, and health workers were generally satisfied with the intervention components. Using the CFIR framework, we organize our results by factors in the outer and inner settings that either challenged or enabled implementation. We then describe individuals' abilities to engage with the intervention and the components themselves, especially the e-Monitor box. Additionally, we discuss challenges during implementation and factors that facilitate the process and will support ongoing delivery. Table 3 provides an overview of challenges and enablers mapped to the CFIR constructs, a more detailed coding tree can be found in the supplementary files.

## Outer setting

The outer setting represents broad contextual factors that shape the implementation context and create (or inhibit) facilitating conditions for innovation use and engagement. Our interviews with patients and their treatment supporters revealed several factors limiting patient access to TB care, which we have described in detail elsewhere [23]. A health worker summed up the three overarching challenges to TB care in the region as: *"One, the area is highly prevalent of*

**Table 3. Implementation challenges and enablers presented by CFIR construct.**

| Construct | Theme | |
|---|---|---|
| | *Challenges* | *Enablers* |
| *Outer setting:* | • *Factors that limit patient access to TB care* | • *Social structure of large families living together enable multiple treatment supporters to take on different roles* |
| *Inner setting:* | • *TB policy at the highest levels affect innovation implementation*<br>• *Recent TB service delivery reforms created a transition period where roles and responsibilities were unclear* | • *Proximity to village doctors and strong links between township doctors and village doctors supports innovation delivery*<br>• *Free TB treatment from the government enables retention in TB care* |
| *Individuals and innovation:* | • *Older patients face digital literacy and language barriers to using WeChat*<br>• *The e-Monitor box design is not convenient for younger patients*<br>• *Health workers have limited time to use WeChat* | • *Mobile phones enable communication in ways beyond the intervention components*<br>• *HCWs teach patients about WeChat and providing support for the e-Monitor*<br>• *e-Monitor alarm function enables timely medication taking* |
| *Implementation process:* | • *Timing of training and gaps in HCW training on innovation components shape innovation delivery* | • *The innovation can be further embedded when used as a tool for broader education on TB in the region*<br>• *Patients, treatment supporters, health workers, and community members can be leveraged to champion the intervention* |

*tuberculosis; Two, the area does not have the best healthcare environment; and three, the public has no understanding of tuberculosis,"* [T55_Township doctor_Tingri]. These challenges shape barriers identified by patients included physical access challenges, out-of-pocket costs, medication side effects, and concerns about treatment effectiveness; while enablers to TB care included knowledge or past experience with TB, integrated care models, supportive village doctors, free TB treatment, subsidies, and family and social support [23].

**Inner setting**

The inner setting relates to health system factors that shape innovation use including health system organization and characteristics of implementation sites, as well as the how TB care is provided. In Shigatse, TB policies and the uncertainty in roles and responsibilities during the transition period following recent TB service delivery reforms were the main challenges reported. Despite these challenges, the role of village doctors within the health system was reported as a key enabler to intervention implementation. In addition, COVID-19 shaped the inner setting and continued to affect the delivery of TB care in the region. For example, during the COVID-19 pandemic, the home-visit DOT strategy was disrupted in some months due to isolation policies. However, patients receiving the intervention maintained patient management through remotely monitored medication boxes and WeChat-based communication.

Our interviews with providers revealed how limited implementation of the Chinese National Tuberculosis Plan in Tibet has shaped the inner setting and affected innovation implementation, which has been described elsewhere [38,39]. Health workers and CDC staff emphasized that TB programs and improving the quality of TB care must become a top policy priority in Shigatse for the intervention to succeed and be sustained over time. More broadly, one CDC Staff emphasized that efforts must be made to break down the silo-ed approach to TB service delivery from the highest levels, *"I think we need to increase awareness and promote inter-department cooperation, like [we have done for] COVID-19, find all the patients and provide proper responses…"* [T62_CDC Staff_Samzhubze]. Recent reforms have been the first step towards greater alignment with the NTP and the intervention represents a concerted effort to improve the quality of TB care across Shigatse through more team-based TB care coordinated through township and county hospitals.

Yet, health workers and CDC staff also reported how recent TB service delivery reforms in the region created a transition period where roles and responsibilities for TB care were unclear. As one senior CDC staff explained

*"The CDC started handing over the TB prevention responsibility to the hospitals in June 2019. After the handover, the whole TB prevention efficiency and quality dropped, because although we organized orientations at the hospitals, many details in the TB prevention process were neglected… the communication and transition between the CDC, township hospitals, as well as village clinics lacked clarity and efficacy at first due to unfamiliarity of the newly divided responsibilities. Now we are starting to get back on track again,"* [T62_CDC Staff_Samzhubze].

The learning curve associated with the reforms was also seen in the intervention. Early-on some health workers were unclear on their scope of duties and followed a more 'traditional' and hierarchical model rather than a team-based care approach with village health workers. Overall though, the role of village doctors was emphasized as a key feature of the health system that supported intervention implementation and strengthened the circle of care [38]. Their proximity and close relationships with communities, as well as their strong links with township doctors enabled village doctors to effectively deliver intervention components. As one county hospital doctor explained, *"Explain to the patients about the advantages of e-Monitor box. If the explanation isn't clear, go to the grassroot level, let village doctors in the village supervise and encourage them. The village doctors can supervise during meetings, the village doctors play a very important part,"* [T41_County hospital doctor_Sa'gya]. Supporting village doctors to safely provide high quality TB care requires ongoing training that acknowledges their crucial role in linking communities to care and that further establishes them as members of a multidisciplinary TB care team.

## Individuals and innovation

The individuals category of the CFIR describes the roles and characteristics of individuals who engage with the innovation and technology-oriented factors that shape innovation use including ease of use and complexity of the technology.

Overall people were satisfied with the intervention and able to engage with the WeChat app and eMonitor box. Most health workers reported that the intervention could improve patient care, and all patients and treatment supporters reported that engaging in the intervention was beneficial. As one treatment supporter reported *"Before, it was the patient's brother that ensures that she takes the medications, after the distribution of e-Monitor box, the patient takes the medications herself,"* [T42_Family treatment supporter_Tingri]. Older patients specifically highlighted the usefulness of the alarm feature of the e-Monitor box and all patients contacted the village doctor or township doctor to successfully troubleshoot transient issues that arose. One 64-year-old patient illustrated the ease of using the e-Monitor box by explaining that *"I can't read. [My phone] can only be used to make calls… I use the [e-Monitor] pill box every day. It can remind me to take medicine…no one [helps me] I am using it myself,"* [T34_Patient_Sa'gya).

However, participants faced various individual-level challenges and barriers in engaging with the text-based WeChat parts of the intervention, largely shaped by language, literacy, and digital literacy. These did not impact e-Monitor use but shaped how participants engaged with their circle of care.

Language and literacy challenges included limited digital literacy as well as limited written literacy and differing levels of literacy between written and spoken Mandarin and Tibetan. Many patients and treatment supporters older than 40 reported only using their phones for voice calls and not using WeChat at all. For others, language barriers prevented them from using WeChat with health workers who write in Chinese characters, as one patient explained *"I make phone calls, I will also watch videos on my phone. I don't send text messages because I cannot recognize Chinese characters. I type in Tibetan in WeChat,"* [T12_Patient_Samzhubze]. From the health worker perspective, some township hospital doctors reported language barriers between village doctors and patients that limited their use of WeChat for communication. As one township doctor explained, *"Some village doctors also don't know how to use WeChat to communicate with patients, it is a bit more difficult. They can understand Chinese characters but can't speak Mandarin."* [T09_Township hospital doctor_Sa'gya]. In Shigatse, eHealth interventions need to be flexible to ensure all literacy and language levels are able to engage with the core components of the intervention.

To overcome these barriers, participants and health workers adapted the intervention through alternative communication methods. The most common adaptation to the intervention was to use phone calls instead of WeChat, followed by house visits, to check in on patients. These adaptations still allowed for patients and treatment supporters to engage more regularly with health workers. As one village doctor described, *"WeChat is used relatively less, because patients don't have WeChat, if there's a problem, we usually go to their door for follow-up or follow-up through the phone,"* [T15_Village doctor Samzhubze]. Some patients have reported relying on family members to help them use technology [23]. For some participants, younger family members were integral in facilitating WeChat use or using WeChat on behalf of the patient. As one 56-year-old patient illustrated, *"Usually 14 people live in the house…My family members remind me, usually it will be my wife or mother reminding me. My wife is my supporter. But because we do not use WeChat, our daughter-in-law became the supporter,"* [T02_Patient_Samzhubze]. However, these social structures are shifting, and more often younger generations are migrating to urban centres for work and school. A township hospital doctor described how this can impact the intervention, *"If the WeChat [account] we 'befriended' belongs to the family members, many family members go out for work, they do not understand the patient's true condition,"* [T26_Township hospital doctor_Sa'gya]. In these cases, the intervention may need to be adapted with closer follow up by the village doctor.

Some township hospitals doctors reported that instructing village doctors and patients on how to use WeChat became a part of their role in delivering the intervention. A few, however, reported that teaching WeChat took time away from other activities. As one township doctor explained, *"Many patients don't know how to use WeChat, we can teach the patient how to use WeChat through health education and promotion…If they went to our base, we would teach the patient and the*

*accompanying village doctor how to use WeChat one-on-one…In total [it takes] about one hour, it takes longer to set up WeChat, takes about half an hour from account registration to start of usage"* [T27_Township hospital doctor_Sa'gya].

Some groups faced challenges more difficult to overcome, including students and government workers. Younger age groups easily used WeChat, but the e-Monitor itself created a barrier to adherence given its size and visibility, as it is a bright blue box. One township doctor described specific problems faced by students, *"Students go to school in Mainland China, and because of many inconveniences, will return the pillbox, and then after the holidays, will take the pillbox back home and continue using it…It isn't convenient to carry it at school and there are problems with plugging it in at school. He takes the tuberculosis medication in secret as he does not want his classmates to know that he has tuberculosis,"* [T27_Township doctor Sa'gya]. A county hospital doctor expanded on groups who may not be able to use the e-Monitor explaining how *"Students, drivers, government employees think that the e-Monitor box is too big and inconvenient to carry, so they are reluctant to use it…The project of electronic medicine box is good, but students and herdsmen will think it is too big and inconvenient to carry without carrying it,"* [T41_County hospital doctor_Sa'gya].

## Implementation process

The implementation process describes the activities and strategies used to implement the innovation. The timing of training was a challenge to effective implementation and gaps in health worker training on the intervention components shaped its delivery. The implementation process, however, presents an important opportunity for broader education on TB in the region, both for health workers and the general population.

Gaps in health worker knowledge on the intervention were attributed to the time delay between training on the intervention and first patient enrolment and the need for self-directed learning to operate the e-Monitor box system. One township doctor described their situation as *"We had our first patient two months after training, we also had to discover how to operate the system by ourselves based on the operation manual. I had forgotten about the section on village doctors, and I probably did not read the village doctor section of the operation manual... I have read the training supervising manual for township hospital doctor, it is okay, I can understand the content,"* [T22_Township hospital doctor_Gyangze].

Others emphasized the need for greater detail in the training sessions and ongoing engagement with the intervention designers for refresher training as the intervention rolled out. As a county hospital doctor explained *"The training content is sufficient; but some places regarding the e-Monitor box are a bit vague and can't be understood; training materials were distributed; no discussions were organized after training, but we [health worker]) discuss about them during treatment,"* [T39_County hospital doctor_Tingri]. Health workers had ideas for improvement including holding joint training with different health workers and participating in mock scenarios. A township hospital doctor reported that *"I think, when conducting training, it would be better to make village, township hospital, and [county] hospital doctors all attend, sometimes we forget things when supervising the village doctors. It would be better to do mock practices…All project participants should participate in the training together, village doctor, township hospital doctor, country hospital doctor. Work tasks should have model case scenarios, live operation [of the e-Monitor], and actual operation process,"* [T22_Township hospital doctor_Gyangze]. While others suggested to have training twice a year to ensure health workers understood their roles and responsibilities in implementing the intervention.

Some health workers highlighted how the current training curricula could be an opportunity to address limited knowledge of TB amongst health workers in the region. Training could then introduce important concepts such as multi-drug resistant tuberculosis and new tools for diagnosis and treatment to all levels of health worker. A township hospital doctor explained that *"The orientation is somewhat lacking in lecturing relevant information on tuberculosis, such as what is tuberculosis, how to manage tuberculosis, how to diagnose suspected patients and so on…Usually training at the municipal and autonomous regions is only for county-level staff. If possible, the usual training can incorporate staff at the township level, as some knowledge is not transmitted to the township and village levels,"* [T26_Township doctor_Sa'gya]. Some village doctors similarly called for increased training on TB to be built into the orientation with one calling for

implementers *"To teach more basic knowledge such as basic vaccination and diagnosis..."* [T29_Village doctor_Sa'gya]. Others suggested that training must be done in Tibetan language to ensure all health workers have adequate knowledge of TB, particularly village doctors who are often the first point of patient contact with the health system. Participants also emphasized the role of the government in ensuring that documentation and processes are in place for this training and to support health workers more broadly. As one township hospital doctor reported, *"The government has to put a strong emphasis on this task [TB care] and publish some relevant paperwork and allow for the active screening of the disease. Since patients are less motivated to be actively seeking medication attention, medical and clinical institutions should organize more orientations for the doctors on the use of medications,"* [T26_Township doctor_Sa'gya]. Strengthening TB care in the region, and improving the quality of care delivered, requires an adequately trained health work force beginning at the most local levels with a shared understanding of TB and how to identify, refer, diagnose, and treat people presenting with the symptoms of TB.

Some health workers offered strategies to increase intervention uptake, address specific concerns, and promote intervention and TB awareness amongst patients. In Sa'gya, health workers described how some patients were initially hesitant about the e-Monitor but that health education about TB and the goal of the intervention can increase their interest in using the e-Monitor. As one township doctor explained *"Put the emphasis on health education, you should promote and educate the main function of the e-Monitor box more, the main goal of the e-Monitor box, if the patients knew about these information they'd be more willing to accept the e-Monitor box,"* [T27_Township doctor_Sa'gya]. A village doctor explained how implementation can be strengthened by engaging more deeply at the most local levels, *"From the perspective of advertisement, village health officials should promote it to patients who easily forget to take medicine. Other villagers also need to know about this project, so the project will be more acceptable if there is another patient,"* [T20_Village doctor_Samzhubze]. Another suggested that, *"It would be better for the village head to come in and persuade the patients, it is easier this way,"* [T61_Village doctor_Tingri]. These perspectives underscore the importance of engaging communities broadly in the intervention and the importance of community opinion to influence uptake and use. To this end, patients can also be champions for the intervention and to spread TB awareness, as one patient explained, *"Sometimes when I encounter patients with symptoms like mine, I will persuade them to go to the township or county hospitals to check and diagnose the disease. I persuade them for early detection and early treatment,"* [T35_Patient_Sa'gya]. Similarly, treatment supporters who had existing awareness of TB strongly supported the intervention and perceived its benefits for their family and others. To ensure intervention sustainability, awareness and engagement at all levels is key, while also addressing broader awareness of TB through better education and health worker training.

## Discussion

This process evaluation explored how contextual, individual, and process-related factors shaped the implementation of an eHealth intervention to strengthen the circle of TB care in Shigatse, Tibet. Guide by the CFIR framework, we identified factors that influenced program delivery with implications for program sustainability.

### Adaptations and user experience

Our findings confirm that eHealth interventions require flexibility to accommodate and adapt to diverse user needs. Implementers should anticipate adaptations of eHealth components as users find ways to create their circle of care given their unique settings and circumstances. In our setting, older patients generally engaged well with the e-Monitor and many used it independently despite limited literacy. However, language and digital barriers constrained use of WeChat leading to adaptations such as phone calls or home visits to maintain communication. These adaptations persevered the intent of the WeChat component of the intervention but, similar to other studies of e-Health interventions in China, underscore the need for anticipatory planning and human-centered design to ensure modifications support intervention goals [40]. Conversely, younger users, including students and the mobile workforce, found WeChat easy to use but considered the

e-Monitor inconvenient due to its size and visibility, which undermines adherence monitoring, a core component of the intervention. Similar challenges have been reported in other settings, further highlighting the importance of differentiated strategies for specific populations [20,41]. Needs assessments and iterative design approaches can help implementers anticipate such scenarios and develop solutions that minimize disruption to intervention fidelity, or that ensure adaptations support the intended intervention goals [42].

## Training for all health workers

Successful implementation, and indeed successful adaptation, requires timely and comprehensive training for involved health workers on all components of the intervention and how these components relate to project goals [20]. Delays between intervention training and patient enrollment, coupled with variability in content across health worker levels, can amplify gaps in knowledge and confidence when caring for people living with TB [43,44]. In or prior work we found low awareness of TB amongst health workers in the region, and in the work described here participants called for refresher sessions on the intervention and joint training across professional categories to bridge this gap [38]. Strengthening overall TB knowledge during these sessions could address broader gaps in awareness in Shigatse while reinforcing the intervention's objectives. Similar findings from other high-burden settings suggest that embedding capacity-building within implementation strategies is essential for sustainability [45]. WeChat may offer an additional platform for ongoing health worker training and delivering TB education in rural and remote settings in Tibet and elsewhere in China [46].

## Implementation science for health policy and system improvement

Intervention implementation must be integrated with, and offer insights on how to achieve, broader TB goals and health policy objectives even in the face of health service disruption [47,48]. For example, recent TB service delivery reforms in the region created transitional challenges but also opportunities to integrated digital adherence technologies into routine care [27,49]. Understanding intervention implementation offered insights on how to provide more people-centered TB care, increase access to TB health services for remote populations, raise greater awareness of TB symptoms and screening, align health workers on TB care delivery protocols, and better embed eHealth in TB health service delivery. For example, the COVID-19 pandemic disrupted TB health services in Tibet for some months due to public health and social measure policies, yet the intervention supported ongoing care delivery through remote monitoring during this time. In this way, framework and theory-guided implementation science is not only a driver of practice and health system improvements but also offers actionable insights for decisionmakers determining health policy. Future research should closely engage with decisionmakers to ensure implementation insights strengthen the delivery of high-quality TB care. For example, recent multi-country studies have evaluated the implementation of digital adherence technologies, including Medication Event Reminder Monitors (MERM), Video Directly Observed Therapy (VDOT), and 99DOTS, highlighting both barriers and facilitators to their use. A scoping review using the RE-AIM framework found that while pillbox and video-based DATs had higher engagement rates, their reach was often constrained by cellphone access and technological challenges [50]. Similarly, the ASCENT trial conducted in the Philippines, South Africa, Tanzania, and Ukraine emphasized the importance of differentiated care models and found that effectiveness of digital adherence technologies varied by setting, with acceptability and feasibility shaped by local infrastructure and user needs [51].

## Strengths and limitations

This manuscript is strengthened by the perspectives of patients, treatment supporters and health workers in Shigatse, Tibet. A further strength is our use of CFIR framework to guide this process evaluation and identify challenges and enablers to intervention implementation. To our knowledge, this is the first study to explore the implementation process of an eHealth program to strengthen TB care in Shigatse. This work was further strengthened by our use of semi-structured interviews in both Tibetan and Mandarin, which enabled participants to describe their experiences in greater detail.

A key limitation of the study is social desirability bias, where participants may have presented a more positively framed account of their engagement with the intervention. We attempted to limit this bias by reassuring participants that the study was not related to their TB care or work performance, and that all responses would be kept confidential. The impact of this bias was likely minimal, as participants were candid in sharing their perspectives, including negative ones. We also acknowledge selection bias, as participants were recruited through the clinical trial, which may have excluded individuals with differing experiences or levels of engagement. Additionally, translation and interpretation bias may have influenced the findings, as interviews conducted in Tibetan and Mandarin were translated into English, potentially losing cultural or technical nuances in the process, an idea which we have explored in our collaborative autoethnography on the translation process [32]. Although we tried to recruit a breadth of participants, vast travel distances and limited resources may have further constrained the diversity of perspectives, which could limit our interpretation of the implementation process.

While this study is grounded in the unique sociocultural and geographic context of Shigatse, several findings, such as the importance of tailoring digital tools to user needs, ensuring comprehensive health worker training, and adapting interventions to local infrastructure, are transferable to other remote, resource-limited settings. However, some cultural or health system aspects may be more specific to the Tibetan context or context of other rural settings in China and should be carefully considered when adapting similar interventions elsewhere. Reflecting on equity, future implementations must ensure that digital health tools are inclusive and accessible across age, education, and linguistic groups to avoid reinforcing existing disparities.

## Conclusion

Using the CFIR framework, this qualitative process evaluation identified several challenges and enablers that shape the implementation of a multi-component eHealth intervention to strengthen the circle of TB care in Shigatse, Tibet, China. Importantly, we highlight intervention adaptations, the ways in which these meet user needs, and how some adaptations support project goals, while others can undermine them. Based on these results, we offer key takeaways for TB eHealth intervention implementation that underscore the importance of infrastructure in creating a consistent user experience, the need to anticipate and document adaptations across user groups, the role of training as a way to strengthen intervention role while capacity-building health workers, and the need for implementation science to feed into health policy objectives. This work highlights the importance of using implementation science to better understand the implementation context and the experiences of users engaging with interventions in practice, particularly in resource-constrained settings.

## Supporting information

**S1 Checklist. Consolidated criteria for reporting qualitative studies (COREQ): 32-item checklist for each study.**
(DOCX)

**S1 Fig. Simplified code tree.**
(TIF)

## Acknowledgments

The authors would like to acknowledge all members of the trial and the support teams and staff at the respective institutions for their contributions to this study. We would like to thank the participants for their cooperation and sharing, the data collectors for their efforts, and the translation team for their dedication.

## Author contributions

**Conceptualization:** Victoria Haldane, Pande Pasang, Jun Hu, Xiaolin Wei.

**Data curation:** Zhitong Zhang, Tingting Yin, Bei Zhang, Yinlong Li, Qiuyu Pan.

**Formal analysis:** Victoria Haldane, Zhitong Zhang.

**Funding acquisition:** Xiaolin Wei.

**Investigation:** Zhitong Zhang, Tingting Yin, Bei Zhang, Yinlong Li, Qiuyu Pan.

**Methodology:** Victoria Haldane.

**Project administration:** Zhitong Zhang, Pande Pasang, Jun Hu.

**Supervision:** Xiaolin Wei.

**Validation:** Xiaolin Wei.

**Writing – original draft:** Victoria Haldane.

**Writing – review & editing:** Victoria Haldane, Zhitong Zhang, Tingting Yin, Bei Zhang, Yinlong Li, Qiuyu Pan, Katie N. Dainty, Elizabeth Rea, Pande Pasang, Jun Hu, Xiaolin Wei.

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
