## [Decision Letter · Decision Letter 0]

25 Jun 2025

PGPH-D-25-00906

A process evaluation of an eHealth intervention to strengthen the circle of tuberculosis care in Shigatse, Tibet, China

Dear Dr. Wei,

Thank you for submitting your manuscript to PLOS Global Public Health. After careful consideration, we feel that it has merit but does not fully meet PLOS Global Public Health’s publication criteria as it currently stands. Therefore, we invite you to submit a revised version of the manuscript that addresses the points raised during the review process.

This is an important topic that has gained prominence in recent years. The reviewers' comments cover a variety of topics, from methodological to thematic issues, with relevant suggestions for improving the work.

We look forward to receiving your revised manuscript.

Kind regards,

Dione Benjumea-Bedoya, Ph.D

Guest Editor

Journal Requirements:

Additional Editor Comments (if provided):

Reviewers' comments:

Reviewer's Responses to Questions

**Comments to the Author**

1. Does this manuscript meet PLOS Global Public Health’s publication criteria?

Reviewer #1: Partly

Reviewer #2: Yes

2. Has the statistical analysis been performed appropriately and rigorously?

Reviewer #1: Yes

Reviewer #2: N/A

3. Have the authors made all data underlying the findings in their manuscript fully available (please refer to the Data Availability Statement at the start of the manuscript PDF file)?

Reviewer #1: No

Reviewer #2: Yes

4. Is the manuscript presented in an intelligible fashion and written in standard English?

Reviewer #1: No

Reviewer #2: Yes

Reviewer #1: Your manuscript tackles an important and under-researched question: how multi-component digital adherence technologies can be implemented for TB control in a geographically remote, high-burden setting. Framing the study with the CFIR and reporting against COREQ demonstrates methodological intent, and the findings are immediately useful for TB programme managers. Nevertheless, several issues must be addressed to meet PLOS Global Public Health standards.

INTRODUCTION

1. Provide a brief rationale for choosing CFIR over alternative frameworks.

METHODS

1. Although it is mentioned that interviews ceased when no new codes emerged, it is suggested to describe how saturation was determined (operational criteria, evidence that the last 3-5 cases did not contribute new codes).

2. Although it is mentioned that two analysts developed and refined the coding framework and resolved discrepancies through discussion. It is suggested to include examples of the code tree or a coded excerpt to illustrate the application of CFIR (and comply with item 25 of COREQ which appears as "N/A")

3. Justify sample size and provide numbers approached / refused (and comply with item 13 of COREQ which appears as "N/A")

RESULTS

I recommend reporting the number of participants approached, refused, or excluded, and reasons where known.

DISCUSION

1. The manuscript briefly alludes to UTAUT framework to justify the relevance of digital adherence technologies, but in the Methods section and throughout the analysis — only uses CFIR. This isolated mention creates a "framework gap" that weakens the theoretical-methodological coherence.

2. Consider referencing recent multicountry studies that have evaluated both barriers/facilitators and implementation outcomes of digital adherence technologies for TB—such as MERM, VDOT, and 99DOTS. Linking your findings to this broader literature will strengthen the discussion and highlight how your results compare with other implementation experiences.

3. Acknowledge selection bias (participants recruited via the clinical trial) and possible social-desirability bias. It is suggested to also acknowledge Translation/interpretation bias, as cultural or technical nuances may be lost between Tibetan, Mandarin, and English.

4. It is suggested to reflect on equity and transferability. What aspects are specific to the Tibetan context and which can be generalized to other remote, resource-limited settings?

5. The CFIR framework serves as a determinants model that systematically organizes barriers and facilitators affecting intervention implementation. However, CFIR alone does not define or assess Implementation Research Outcomes (IROs). To evaluate how well something was implemented, CFIR should be paired with outcome-focused frameworks, such as the widely-used Implementation Outcomes Framework (Proctor et al., 2011). The Discussion and Conclusion sections state that the intervention was “accepted” and “sustainable,” yet no IROs were measured to substantiate these claims. If the study’s explicit aim was solely to identify CFIR‐based determinants, this methodological boundary should be clearly stated in the Methods as a deliberate scope limitation. Consequently, the Discussion and Conclusions must remain aligned with the data actually collected, focusing on barriers and facilitators, and avoid drawing inferences about implementation outcomes that were not evaluated.

LANGUAGE AND PRESENTATION

The manuscript is generally intelligible but needs systematic copy-editing. Typical issues include:

- Spelling mistakes “Directory Observed Therapy” : correct (Directly Observed Therapy)

- Repeated misspelling “Organizaton” : correct (Organization)

- Long sentences (>40 words) in Discussion—split for readability.

- Acronyms: present them in full the first time and maintain the same form ("DOTS" vs. "DOT")

- Maintain capitalization consistency (“e-Monitor” v. “e-monitor”)

- Verb tenses: in Results use consistent simple past ("participants reported" instead of alternating with present).

- Use of quotation marks: standardize single/double quotes

- "There are some incomplete sentences, for example: "Two reviewers (VH, ZZ) … discussed how the CFIR constructs.".

Reviewer #2: Overall, this is a study that demonstrates rigor and ethical responsibility; however, it has aspects that require further development. It is important to address how it was decided to work only on some CFIR constructs, as well as to review the UTAUT framework, which will be mentioned only at the end of the document.

The structure of the document is consistent with the requirements of the journal. The different sections reflect the expected content. However, the information available can be further leveraged to achieve solid results that can be useful for patients, individuals, and healthcare institutions.

The analysis plan requires an explanation of the process followed to arrive at the categories. In fact, no clear categories or subcategories that emerged from the process are addressed; only a more deductive than inductive analysis process is suggested. Some important elements are identified in the testimonies (see comments in the manuscript) that may lead to emerging findings of great value for the process.

Ethical considerations require clarity in the treatment of information from participants under 18 years of age, as the consent of parents or legal guardians is not sufficient.

In the discussion, it is important to analyze the results in light of other studies and health policies that demonstrate the relevance of these results for the implementation of local, national, and global public policies, as well as the opportunity and need to develop new policies.

In the attached manuscript you can see detailed observations that may be of interest to you in making adjustments to it.

**Do you want your identity to be public for this peer review?** For information about this choice, including consent withdrawal, please see our Privacy Policy

Reviewer #1: No

Reviewer #2: No

---

## [Decision Letter · Decision Letter 1]

18 Sep 2025

PGPH-D-25-00906R1

A process evaluation of an eHealth intervention to strengthen the circle of tuberculosis care in Shigatse, Tibet, China

Dear Dr. Wei,

Thank you for submitting your manuscript to PLOS Global Public Health. After careful consideration, we feel that it has merit but does not fully meet PLOS Global Public Health’s publication criteria as it currently stands. Therefore, we invite you to submit a revised version of the manuscript that addresses the points raised during the review process.

eHealth interventions are valuable in strengthening care for patients with tuberculosis. Please take into account all reviewers' comments, as some of the responses to this review have not been clearly included in the manuscript.

We look forward to receiving your revised manuscript.

Kind regards,

Dione Benjumea-Bedoya, Ph.D

Guest Editor

Journal Requirements:

Additional Editor Comments (if provided):

Reviewer #1:

Reviewer #2:

Reviewers' comments:

Reviewer's Responses to Questions

**Comments to the Author**

Reviewer #1: All comments have been addressed

Reviewer #2: All comments have been addressed

publication criteria?

Reviewer #1: Yes

Reviewer #2: Yes

3. Has the statistical analysis been performed appropriately and rigorously?

Reviewer #1: N/A

Reviewer #2: N/A

4. Have the authors made all data underlying the findings in their manuscript fully available (please refer to the Data Availability Statement at the start of the manuscript PDF file)?

Reviewer #1: No

Reviewer #2: No

5. Is the manuscript presented in an intelligible fashion and written in standard English?

Reviewer #1: Yes

Reviewer #2: Yes

Reviewer #1: Thank you for submitting the revised version of your manuscript. I have analyzed the implemented changes and the response letter in detail, and I find that you have satisfactorily addressed all the suggestions made. The manuscript has improved considerably, especially in its methodological rigor and the contextualization of the findings within the existing literature. My following evaluation focuses on the implementation of these recommendations and offers a final critical perspective on a few aspects that could further strengthen the impact of your work.

1. A slight discrepancy has been identified regarding the description of data saturation. While the criterion has been rightly added to the Methods section, the COREQ checklist (S1 File) refers to the Results section for this point. Please ensure that the information in the checklist is harmonized to match the main text.

2. The manuscript mentions that COVID-19 "shaped the inner setting" and will continue to affect the delivery of TB care. However, this idea is not further developed. Given that the recruitment period extended to the end of 2020, the pandemic was an unavoidable contextual factor. It would be enriching to add a sentence or two in the "Discussion" section reflecting on how the pandemic might have interacted with the intervention.

3. In the "Implementation Process" section, it is suggested that patients, family members, and community leaders can be "champions" to promote the intervention and TB awareness. This is a key implementation strategy. In the "Discussion" section, the importance of formalizing this role in future scale-up phases could be briefly mentioned. This would transform an interesting observation into a more solid and actionable programmatic recommendation.

Reviewer #2: The authors have been clear in their responses to the reviewers' concerns and recommendations; however, while the reviewers' responses to comments are sometimes perceived as clear, they are not always as clear in the manuscript. I emphasize that the comments in the initial manuscript could contribute to improving the document. I mention one aspect as an example: it is clear that no minors under 18 years of age were interviewed, but for ethical reasons, it is understood that minors under 18 years of age did participate.

I recommend reviewing each of the answers and making sure that they are effectively reflected in the manuscript.

**Do you want your identity to be public for this peer review?** For information about this choice, including consent withdrawal, please see our Privacy Policy

Reviewer #1: No

Reviewer #2: No

---

## [Decision Letter · Decision Letter 2]

17 Nov 2025

PGPH-D-25-00906R2

A process evaluation of an eHealth intervention to strengthen the circle of tuberculosis care in Shigatse, Tibet, China

Dear Dr. Wei,

Thank you for submitting your manuscript to PLOS Global Public Health. After careful consideration, we feel that it has merit but does not fully meet PLOS Global Public Health’s publication criteria as it currently stands. Therefore, we invite you to submit a revised version of the manuscript that addresses the points raised during the review process.

EDITOR:

The results of this e-health intervention are important in the field of tuberculosis; however, one of the reviewers pointed out some important details that need to be addressed to improve the work.

We look forward to receiving your revised manuscript.

Kind regards,

Dione Benjumea-Bedoya, Ph.D

Guest Editor

Journal Requirements:

Please include a complete copy of PLOS’ questionnaire on inclusivity in global research in your revised manuscript. Our policy for research in this area aims to improve transparency in the reporting of research performed outside of researchers’ own country or community. The policy applies to researchers who have travelled to a different country to conduct research, research with Indigenous populations or their lands, and research on cultural artefacts. The questionnaire can also be requested at the journal’s discretion for any other submissions, even if these conditions are not met. Please find more information on the policy and a link to download a blank copy of the questionnaire here: https://journals.plos.org/globalpublichealth/s/best-practices-in-research-reporting. Please upload a completed version of your questionnaire as Supporting Information when you resubmit your manuscript.

Additional Editor Comments (if provided):

The results of this e-health intervention are important in the field of tuberculosis; however, one of the reviewers pointed out some important details that need to be addressed to improve the work. Authors are encouraged to address the reviewers' comments.

Reviewers' comments:

Reviewer's Responses to Questions

**Comments to the Author**

Reviewer #1: All comments have been addressed

Reviewer #2: All comments have been addressed

publication criteria?

Reviewer #1: Yes

Reviewer #2: Yes

3. Has the statistical analysis been performed appropriately and rigorously?

Reviewer #1: N/A

Reviewer #2: N/A

4. Have the authors made all data underlying the findings in their manuscript fully available (please refer to the Data Availability Statement at the start of the manuscript PDF file)?

Reviewer #1: Yes

Reviewer #2: No

5. Is the manuscript presented in an intelligible fashion and written in standard English?

Reviewer #1: No

Reviewer #2: Yes

Reviewer #1: This revised manuscript represents an improvement over prior versions. The authors have addressed the majority of reviewers’ previous comments and aligned the paper with PLOS Global Public Health’s expectations. Minor editorial and conceptual adjustments are the only remaining issues before acceptance:

1. Although the COREQ checklist and S2 File indicate two coders, please add a line in the Methods describing how coding discrepancies were resolved.

2. The Discussion section is overly detailed and extends beyond the recommended length for PLOS Global Public Health. Some paragraphs, particularly those describing user experiences with the e-Monitor and WeChat, reiterate similar points about accessibility and adaptation. These sections could be streamlined or synthesized to avoid redundancy, improve narrative flow, and highlight the most salient analytical insights.

3. Additionally, some minor writing errors that were previously flagged remain uncorrected, including inconsistent terminology (e.g., “Directly Observed Therapy” instead of “Director Observed Therapy”) and spelling inconsistencies (e.g., unify “e-Monitor” throughout the text).

Reviewer #2: They have responded to the suggestions and recommendations made by the evaluators in the previous readings.

**Do you want your identity to be public for this peer review?** For information about this choice, including consent withdrawal, please see our Privacy Policy

Reviewer #1: No

Reviewer #2: No

---

## [Decision Letter · Decision Letter 3]

11 Dec 2025

A process evaluation of an eHealth intervention to strengthen the circle of tuberculosis care in Shigatse, Tibet, China

PGPH-D-25-00906R3

Dear Prof Wei,

We are pleased to inform you that your manuscript 'A process evaluation of an eHealth intervention to strengthen the circle of tuberculosis care in Shigatse, Tibet, China' has been provisionally accepted for publication in PLOS Global Public Health.

Best regards,

Dione Benjumea-Bedoya, Ph.D

Guest Editor

Reviewer Comments (if any, and for reference):

Reviewer's Responses to Questions

**Comments to the Author**

Reviewer #1: All comments have been addressed

publication criteria?

Reviewer #1: Yes

3. Has the statistical analysis been performed appropriately and rigorously?

Reviewer #1: N/A

4. Have the authors made all data underlying the findings in their manuscript fully available (please refer to the Data Availability Statement at the start of the manuscript PDF file)?

Reviewer #1: Yes

5. Is the manuscript presented in an intelligible fashion and written in standard English?

Reviewer #1: Yes

Reviewer #1: All the suggestions have now been addressed.

**Do you want your identity to be public for this peer review?** For information about this choice, including consent withdrawal, please see our Privacy Policy

Reviewer #1: No
